# The Measurement Uncertainty in Determining of Electrical Resistance Value by Applying Direct-Comparison Method

Vladimir Pleština *[ID], Vedran Boras and Hrvoje Turić

Faculty of Science, University of Split, Ruđera Boškovića 33, 21000 Split, Croatia; vboras@pmfst.hr (V.B.); turich@pmfst.hr (H.T.)
* Correspondence: vlade@pmfst.hr

**Abstract:** This paper considers the unknown electrical resistance (measurand) as the numerical result of the measurement that was carried out by means of the well-known direct comparison measurement method using an appropriate standard resistor and voltmeter. In the literature, this measurement method is also referred to as a series comparison method. This method of measurement is one of the indirect methods and is suitable for measuring low resistance. This paper presents two approaches for evaluating the unknown electrical resistance and its associated combined standard uncertainty. The entire process of evaluating the combined standard uncertainty that is associated with the measurand and the standard uncertainties that are associated with the analyzed input quantities has been entirely performed in accordance with the applicable international recommendations and guidelines for the uncertainty of measurement. The analyzed approaches for evaluating the combined standard uncertainty are designed to be universal and valid both for the mutually non-correlated input quantities and for the mutually correlated input quantities, which can be obtained from a single observation, or repeated observations or by other means. This paper can substantially contribute to the measurements in electrical engineering and education.

**Keywords:** unknown electrical resistance; uncertainty of measurement; direct comparison measurement; series comparison method

## 1. Introduction

When the result of a quantity measurement is reported, it requires the estimated value of the measurand (the quantity to be measured) and the uncertainty that is associated with that value. The uncertainty of measurement as an attribute for expressing the quality of a measurement result is relatively new in the history of measurement. The acceptance of uncertainty of measurement as a unique numerical expression of the measurement result quality ensued from many years of discussions resulting in international agreements that are outlined in the guidelines [1]. That authoritative document is popularly known as the GUM, (which stands for guide to the expression of uncertainty in measurement). The GUM indicates that the formal definition of "uncertainty of measurement" refers to a parameter that is associated with the measurement result characterizing the dispersion of values that are reasonably attributable to the measurand. Correspondingly, other standards and guides, such as [2,3], are strictly based on the guide.

For many years, the determination and expression of uncertainty in measurement has been subject to debate in several global metrological organizations (IEC, BIMP, ISO, etc.) around the world for many years. Several recommendations, guidelines, and instructions have been generated therefrom. The latest internationally accepted document for the expression of uncertainty in measurement is [2], which was adopted by the European Cooperation for Accreditation (EA).

For many purposes, unknown resistors are compared to standard resistors by a comparison circuit. One such measurement method is the direct comparison measurement

method using a standard resistor and a voltmeter. There are other similar methods, such as the substitution method and the direct comparison method using a standard resistor and potentiometer, to which the same procedure for evaluating and expressing uncertainty in measurement may apply as the procedure that is described in this scientific paper. Thus far, the literature has reported only the numerical value of the unknown resistance that is obtained by means of the direct comparison method, rarely considering the internal resistance of the voltmeter [4–11].

In general, every value that is obtained through measurement has some uncertainty, and even though uncertainty may be reduced by thorough planning, a prudent selection of a measuring instrument, and a careful execution of the experiment, it cannot be eliminated entirely. Therefore, the main goal of this paper is to calculate the value of the unknown electrical resistance using this direct comparison method, as well as to evaluate the associated uncertainty of measurement according to [1,2].

In the proposed model, a measurand (output quantity) $R_X$ is not measured directly, but it is determined from four quantities (input quantities: resistance of a standard resistor $R_N$, input resistance of used voltmeter $R_V$, the voltage drop $U_N$ across resistor $R_N$, and the voltage drop $U_X$ across resistor $R_X$) through a functional relationship $f(\cdot)$, also referred to as the measurement model function. This functional relationship $f(\cdot)$ is based on the theory of a voltage divider. This paper provides a detailed description of the determination of the measurand $R_X$ and its combined standard uncertainty $u_C(R_X)$. Generally, the determination of the combined standard uncertainty of a non-directly measured measurand (output quantity) is described in many textbooks [12–18]. Ref. [19] provides a simplified form of the output quantity $R_X$ evaluation when it is determined from only three input quantities, $R_N$, $U_N$, and $U_X$, by applying the direct comparison measurement method using a standard resistor and voltmeter, in addition to the evaluation of the combined standard uncertainty that is associated with this $R_X$. Unlike the aforementioned research, the model that is proposed in this paper is more comprehensive and analyzed in more detail. The output quantity $R_X$ in this model is determined from either four mutually non-correlated or four mutually independent input quantities, which can be obtained from a single observation, repeated observations, or by other means.

According to [1,2], the steps for evaluating and expressing uncertainty in the measurement of the unknown electrical resistance that is determined according to the proposed method may be summarized as follows:

1. Determination of the functional relationship $f(\cdot)$ (measurement model function) between the unknown electrical resistance $R_X$ (output quantity) and the input quantities, on which it depends. This functional relationship should contain every quantity, including all corrections and correction factors, that can add a significant component of uncertainty to the measurement result.
2. Determination of the estimated value of each input quantity in the relationship, which can be obtained from a single observation, repeated observations, or by other means.
3. Evaluation of the standard uncertainty of each input-estimated quantity. Type A evaluation of the standard uncertainty will be evaluated for an input estimate that is obtained from the statistical analysis of a series of observations, and Type B evaluation of standard uncertainty will be evaluated for an input estimate that is obtained from a single observation or by other means.
4. Evaluation of the covariances that are associated with any correlated input estimates.
5. Calculation of the measurement result.
6. Determination of the combined standard uncertainty of the measurement result from the standard uncertainties and covariances that are associated with the input-estimated quantities that were obtained in step 2.
7. If necessary, the determination of an expanded uncertainty.
8. Presentation of the measurement result.

An illustrative example that is presented at the end of this paper describes the practical application of the proposed measurement method.

The main contributions and originality of the proposed model include:

- the new and the original form of the mathematical expression for the calculation of the electrical resistance value, which is extremely suitable for conducting higher-order partial derivatives which can be of great importance when the model of measurement functions has a nonlinear character,
- the complete the measurement model function,
- the method that is proposed in this paper allows the elimination of the influence of thermo-electrical voltages, if any,
- the proposed model includes the cases where the input quantities are correlated and mutually independent,
- the proposed model can be used in the determination of the electrical resistance value by means of the direct comparison method using a standard resistor and potentiometer, and in case when the unknown resistors are compared to standard resistors by a comparison circuit (comparators).

## 2. Direct Comparison Method Using a Standard Resistor and Voltmeter

In this method, the unknown resistor $R_X$, the standard resistor $R_N$ and a current source are connected in series, as shown in Figure 1, and a voltmeter is used to measure the voltage drop across each resistor.

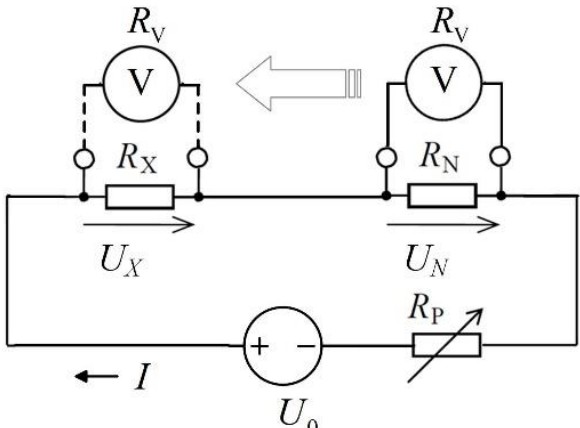

**Figure 1.** Direct comparison method using a standard resistor and a voltmeter.

*2.1. Formulation of the Functional Relationship (Measurement Equation)*

This measurement method is based on the principle of a voltage resistive divider. In this case, a simple example of a voltage resistive divider is presented by two resistors that are connected in a series with the input DC voltage applied across the resistor pair, Figure 2. One of them has the unknown electrical resistance $R_X$ and the second one is a standard resistor with the known electrical resistance $R_N$. According to the principle of a voltage resistive divider, the well-known equation applies:

$$I = \frac{U_X}{R_X} = \frac{U_N}{R_N} \Rightarrow \frac{U_X}{U_N} = \frac{R_X}{R_N} \tag{1}$$

Deriving from (1), the value of the unknown resistance $R_X$ is:

$$R_X = R_N \frac{U_X}{U_N} \tag{2}$$

If the unknown resistance $R_X$ is to be determined by measuring the voltage $U_N$ and $U_X$ using a voltmeter according to Figure 1, then (2) would be valid only if the measurement is performed by using an ideal voltmeter (voltmeter with an infinitely large internal resistance). However, although the voltmeters typically have a large internal resistance $R_V$,

they introduce systematic effects in the measurement of electrical voltage, which in some circumstances may be neglected, and considered in others.

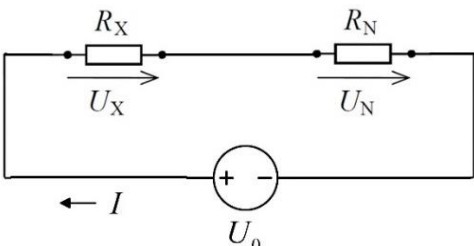

**Figure 2.** Equivalent circuit of a voltage resistive divider with two resistors.

Due to this internal resistance of the voltmeter, a systematic effect occurs in the above voltage measurement, so the measured voltage values $U_X$ and $U_N$ will also contain this systematic effect.

In order to minimize and ignore the systematic effect, the following conditions must be met:

(a)  During the measurement of voltages $U_X$ and $U_N$, the current $I$ in the circuit shown in Figure 1 must remain unchanged, respectively

$$\frac{U_0}{R_N + R_X \parallel R_V} = \frac{U_0}{R_X + R_N \parallel R_V} = I = \text{constant} \tag{3}$$

which is achieved by using a highly stabilized DC power source or by adjusting the variable resistor $R_p$ (see Figure 3).

(b)  During the measurement of voltages $U_X$ and $U_N$, the value of the internal resistance of the voltmeter $R_V$ should remain unchanged, which is achieved if both the measurements are performed within the same voltmeter measuring range.

(c)  The resistance values of $R_X$ and $R_N$ should be approximately in the same order of magnitude and significantly lower (102 to 104 times less) than the voltmeter resistance value $R_V$. Therefore, this measurement method is commonly used when measuring the low resistance values $R_X$ (up to a few k$\Omega$ only).

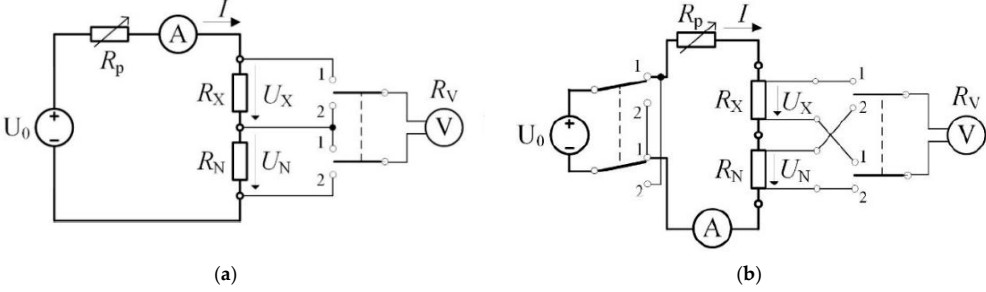

(a)                                               (b)

**Figure 3.** Wiring diagrams that are implemented in practice for the direct comparison method using a standard resistor and voltmeter: (**a**) two-terminal and (**b**) four-terminal connection.

If voltages $U_X$ and $U_N$ are measured simultaneously using the same high-resolution voltmeter while maintaining the same measuring range and without changing the current intensity during the measurement, the systematic effects in measuring these voltages can be neglected (Figure 1). In this case, no correction of the calculated value of the unknown resistance $R_X$ is required, which can be derived from (2), as corroborated by the following observation.

The voltage $U_N$ can be derived from Figure 1 by the following expression:

$$U_N = U_0 \frac{(R_N \parallel R_V)}{R_X + (R_N \parallel R_V)} = I \frac{R_N \cdot R_V}{R_N + R_V} \tag{4}$$

Analogously, the voltage $U_X$ can be derived from Figure 2 by the expression:

$$U_X \;=\; U_0 \frac{(R_X \parallel R_V)}{R_N + (R_X \parallel R_V)} = I \frac{R_X \cdot R_V}{R_X + R_V} \tag{5}$$

Dividing (5) by (4), with the fulfilled conditions (a), (b) and (c) and by using (1), the following equation is obtained:

$$\frac{U_X}{U_N} = \frac{R_X}{R_N} \frac{R_X + R_V}{R_N + R_V} = \frac{R_X}{R_N} \frac{\overbrace{R_X/R_V}^{\to 0} + 1}{\underbrace{R_N/R_V}_{\to 0} + 1} = \frac{R_X}{R_N} \tag{6}$$

It is evident from Equations (2) and (6) that the value of the unknown resistance $R_X$ is independent of the connected voltage $U_0$, the current $I$, or the internal resistance of the voltmeter $R_V$. That means that this method of comparing the voltage with the fulfilled conditions (a), (b), and (c) is exempt from the systematic voltage deviation. Hence, we obtain the following simplified measurement model function:

$$R_X = f(R_N, U_N, U_X) \tag{7}$$

In practice, the following circuit diagrams are commonly used to measure small resistances using the direct comparison method (Figure 3a). The variable resistor $R_p$ adjusts the current intensity $I$ so that it remains unaltered during voltages measurement and lower than the continuous allowed current intensity through the $R_X$ and $R_N$ resistors. That is controlled by means of an ammeter.

The method of four-terminal (four-wire) connections for electrical resistance measurement (according to Figure 3b) is used for precise measurement and for measuring low electrical resistances. The four-terminal (four-wire) connection of both the measured resistor and the standard resistor is the most accurate method when measuring circuits below 10 ohms, as this method eliminates the influence of the terminal resistances and the resistances of connecting wires.

If conditions (a) and (b) are met, but condition (c) is not, the internal resistance of the $R_V$ voltmeter must be considered when determining the unknown resistance $R_X$. Hence, dividing (5) by (4) we obtain:

$$R_X = R_N \frac{U_X}{U_N} \cdot \frac{R_V}{R_V + R_N \left(1 - \frac{U_X}{U_N}\right)} = \left[ \frac{U_N}{U_X} \left( \frac{1}{R_N} + \frac{1}{R_V} \right) - \frac{1}{R_V} \right]^{-1} \tag{8}$$

According to (8), the measurand (output quantity) $R_X$ is not measured directly, but it is determined from four other quantities (input quantities), $R_N$, $R_V$, $U_N$, and $U_X$, through a functional relationship $f(\cdot)$ (measurement model function):

$$R_X = f(R_N, R_V, U_N, U_X) \tag{9}$$

The complete model function (9) is an algorithm that must be evaluated numerically.

The input quantities $R_N$, $R_V$, $U_N$, and $U_X$ upon which the output quantity $R_X$ depends may be viewed as the measurands themselves and depend on other quantities, including corrections and correction factors for systematic effects, thereby leading to a complicated functional relationship $f(\cdot)$ that may never be explicitly noted. No further dependence of the input quantities $R_N$, $R_V$, $U_N$, and $U_X$ on other quantities will be considered in this paper.

## 2.2. Evaluation of the Input Quantities and Their Standard Uncertainties

In order to obtain the numerical value of the unknown resistance $R_X$, according to (8), the measurements of voltage drops $U_N$ and $U_X$ are required, as well as the appropriate value of standard resistor $R_N$.

Each of the input quantities in the model function (9) indicates not only their estimated value, but also their standard uncertainty. With the model function (9), both the output estimated value and the standard uncertainty are then calculated considering the GUM rules.

An estimate of the measurand $R_X$, denoted by $R_{1X}$, is obtained from (9) by using input estimates $R_{1N}$, $R_{1V}$, $U_{1N}$, and $U_{1X}$ for the values of the four input quantities $R_N$, $R_V$, $U_N$, and $U_X$. Hence, the output estimate $R_{1X}$, which is the numerical result of the measurement, is given by

$$R_{1X} = f(R_{1N}, R_{1V}, U_{1N}, U_{1X}) \tag{10}$$

Since the input estimates $R_{1N}$, $R_{1V}$, $U_{1N}$, and $U_{1X}$ in (10) refer to the measurement results, each of them has an associated standard uncertainty $u(R_{1N})$, $u(R_{1V})$, $u(U_{1N})$, and $u(U_{1X})$, which may contribute to the standard uncertainty of the final measurement result $R_{1X}$. The sets of input quantities $R_{1N}$, $R_{1V}$, $U_{1N}$, and $U_{1X}$ are categorized as quantities whose values and uncertainties are directly determined in the current measurement. These values and uncertainties are obtained from a single observation, repeated observations, or by other means, and may involve the determination of corrections of instrument readings and corrections of influence quantities, such as ambient temperature, barometric pressure, and humidity.

In general, the uncertainty of the measurement comprises many components. Some of these components may be evaluated from the statistical distribution of the results of a series of measurements and characterised by experimental standard deviations. The other components, which can also be characterized by standard deviations, are evaluated from the assumed probability distributions based on experience or other information.

In the following section, we will consider the evaluation of the input estimates $U_{1N}$ and $U_{1X}$ and their associated standard uncertainties $u(U_{1N})$ and $u(U_{1X})$, which can be obtained either from the statistical analysis of a series of individual measurements under identical experimental conditions (repeatability conditions)—repeated observations, or from a single observation.

The mean estimate of the supposed distribution of the values is taken as the value of the input quantity, and the estimation of the standard deviation of the mean estimate is taken as the standard uncertainty.

Hence, for the input quantity $U_{1N}$ that is estimated from $n$ independent repeated observations $U_{1N,k}$ ($k = 1, \ldots, n$), the arithmetic mean $\overline{U}_{1N}$ is obtained from equation

$$\overline{U}_{1N} = \frac{1}{n} \sum_{k=1}^{n} U_{1N,k} \tag{11}$$

is used as the input estimate $U_{1N}$ in (10) to determine the measurement result $R_{1X}$; i.e.,

$$U_{1N} = \overline{U}_{1N} \tag{12}$$

Analogously, for the input quantity $U_{1X}$ that is estimated from $n$ independent repeated observations $U_{1X,k}$ ($k = 1, \ldots, n$), the arithmetic mean $\overline{U}_{1X}$ is obtained from equation:

$$\overline{U}_{1X} = \frac{1}{n} \sum_{k=1}^{n} U_{1X,k} \tag{13}$$

is used as the input estimate $U_{1X}$ in (10) to determine the measurement result $R_{1X}$; i.e.,

$$U_{1X} = \overline{U}_{1X} \tag{14}$$

If the input estimates $U_{1N}$ and $U_{1X}$ are obtained from the repeated observations, their associated standard uncertainties are evaluated as Type A evaluation of standard uncertainties $u_A(U_{1N})$ and $u_A(U_{1X})$, respectively. Standard uncertainty $u_A(U_{1N})$ is defined as an estimate of the standard deviation of the distribution of values, termed the experimental standard deviation $s(U_{1N})$. The experimental standard deviation provided a quantitative estimate of the dispersion of the possible measured values $U_{1N,k}$ ($k = 1, \ldots, n$), about their mean value $\overline{U}_{1N}$, and it is given by

$$u_A(U_{1N}) \approx s(U_{1N}) = \sqrt{\frac{1}{n-1}\sum_{k=1}^{n}\left(U_{1N,k} - \overline{U}_{1N}\right)^2} \tag{15}$$

Analogously, the standard uncertainty $u_A(U_{1X})$ is defined as an estimate of the standard deviation $s(U_{1X})$ and is given by

$$u_A(U_{1X}) \approx s(U_{1X}) = \sqrt{\frac{1}{n-1}\sum_{k=1}^{n}\left(U_{1X,k} - \overline{U}_{1X}\right)^2} \tag{16}$$

The best estimate for the standard uncertainty $u_A(U_{1N})$ to be associated with $U_{1N}$ is the experimental standard deviation of the mean and is given by

$$\begin{aligned} u_A(U_{1N}) &= u_A\left(\overline{U}_{1N}\right) = s\left(\overline{U}_{1N}\right) = \\ &= \sqrt{\frac{1}{n(n-1)}\sum_{k=1}^{n}\left(U_{1N,k} - \overline{U}_{1N}\right)^2} \quad (n \geq 10) \end{aligned} \tag{17}$$

The best estimate for the standard uncertainty $u_A(U_{1X})$ to be associated with $U_{1X}$ is the experimental standard deviation of the mean and is given by

$$\begin{aligned} u_A(U_{1X}) &= u_A\left(\overline{U}_{1X}\right) = s\left(\overline{U}_{1X}\right) = \\ &= \sqrt{\frac{1}{n(n-1)}\sum_{k=1}^{n}\left(U_{1X,k} - \overline{U}_{1X}\right)^2} \quad (n \geq 10) \end{aligned} \tag{18}$$

Those input estimates which are not evaluated from the repeated observations must be obtained by other methods, such as those that are indicated in the second category of 4.1.3 in [1].

If the input estimates $U_{1N}$ and $U_{1X}$ are obtained by means of the same voltmeter from a single observation, their standard uncertainties are evaluated as Type B evaluation of standard uncertainties. Presuming that the values of the input estimates $U_{1N}$ and $U_{1X}$ are estimated from an assumed rectangular probability distribution of a lower limit $a_-$ and an upper limit $a_+$, the input estimate $U_{1N}$ is usually the expectation of the rectangular probability distribution

$$U_{1N} = \frac{a_+ + a_-}{2} \tag{19}$$

while the standard uncertainty $u_B(U_{1N})$ to be associated with $U_{1N}$ is the positive square root of the distribution variance

$$u_B(U_{1N}) = \frac{a_+ - a_-}{2\sqrt{3}} \; (\text{V}) \tag{20}$$

If $a_+ = -a_- = a$ then

$$U_{1N} = a \tag{21}$$

and

$$u_B(U_{1N}) = \sigma = \frac{a}{\sqrt{3}} \; (\text{V}) \tag{22}$$

The input estimate $U_{1X}$ can also be obtained by means of (19). Type B evaluation of standard uncertainty $u_B(U_{1X})$ to be associated with $U_{1X}$ can also be derived from (22).

Let us also suppose that the manufacturer accuracy specifications are available for the used voltmeter and that, for the considered range, they provide an interval of possible values, whose half-amplitude $a$ is provided for:

- an analogous voltmeter (such as PMMC voltmeter) by means of:

$$a = \frac{C \cdot M}{100} \text{ (V)} \tag{23}$$

where $C$ refers to the accuracy class of the analogous voltmeter (%), and $M$ to the maximum value of the voltmeter (V) measuring range;

- a digital voltmeter by means of any combination of two or three terms on the right side of the following expression:

$$a = \frac{p_1}{100} X_{DV} + \frac{p_2}{100} M_{DV} + ND \text{ (V)} \tag{24}$$

where

$p_1$—percentage of the voltmeter reading (%),

$p_2$—percentage of the voltmeter range (%),

$X_{DV}$—reading value of the used digital voltmeter (V). In our proposed model $X_{DV} = U_{1N}$ or $X_{DV} = U_{1X}$,

$M_{DV}$—selected measurement range of the digital voltmeter (V), and

$ND$—number of digits, where the lexeme *digit*, sometimes confused with the lexeme *count* due to its similar meaning, indicates the value of the less significant digit for the range in use. The number of digits represents the resolution of the instrument for that range.

Sometimes these combinations also include the absolute value of the measured quantity (volt, ohms, ampere . . . ).

According to the manufacturer of the voltmeter, the interval $U_{1N} \pm a$ encompasses all the values that are reasonably attributable to the estimate $U_{1N}$, hence its coverage probability is, therefore, 100%. In addition, the interval $U_{1X} \pm a$ encompasses all the values that are reasonably attributable to the estimate $U_{1X}$, hence its coverage probability is 100%.

Let us also suppose that the manufacturer's specification is available for the used standard resistor $R_N$. The nominal value of standard resistor $R_N$, that is selected from the manufacturer's specification, will be taken as the input estimate $R_{1N}$. Usually, the nominal values of standard resistors are given with the tolerance band $\pm \Delta R_{N,\max}$ or with the standard resistor tolerance $\delta R_{N,\max}$ (%).

For the input estimate $R_{1N}$, the standard uncertainty is also considered as Type B evaluation of standard uncertainty. Type B evaluation is founded on the assumption of rectangular (uniform) probability distribution and the manufacturer's specification for the selected standard resistor $R_{1N}$. The standard uncertainty $u_B(R_{1N})$ to be associated with $R_{1N}$ is given by

$$u_B(R_{1N}) = \sigma = \frac{\Delta R_{N,\max}}{\sqrt{3}} = \frac{\delta R_{N,\max}/100}{\sqrt{3}} \cdot R_N \text{ } [\Omega] \tag{25}$$

where

$R_N$—nominal value of standard resistor ($\Omega$)

$\delta R_{N,\max}$—tolerance of the standard resistor $R_N$ (%).

Presuming that the manufacturer's specification of the used voltmeter provides the data for the internal resistance of the used voltmeter $R_V$, the nominal value of the input resistance $R_V$ will be taken as the input estimate $R_{1V}$. Since the manufacturer does not provide any additional information about the tolerance (%) of this resistance, apart from the internal resistance of the voltmeter $R_V$, we can ignore the standard uncertainty $u_B(R_{1V})$ to be associated with $R_{1V}$.

This is also permitted in [1], where it has been indicated that in some cases the uncertainty of a correction of a systematic effect does not need to be included in the

evaluation of the uncertainty of a measurement result. Although the uncertainty has been evaluated, it may be ignored if its contribution to the combined standard uncertainty of the measurement result is insignificant, since the internal resistance of the voltmeters is very large in relation to the measured resistance.

### 2.3. Evaluation of Output Quantity and Its Uncertainty

In view of the above, the output estimate $R_{1X}$ is the numerical result of the measurement, and it can be calculated by means of (10), which is obtained from (9) when the unknown input quantities $R_N$, $R_V$, $U_N$, and $U_X$ are replaced by the corresponding estimates $R_{1N}$, $R_{1V}$, $U_{1N}$, and $U_{1X}$ that are obtained from the measurements. According to [1], the output estimate $R_{1X}$ may be obtained by means of the two following approaches:

The output estimate $R_{1X}$ is taken as the functional relationship $f(\cdot)$ from the arithmetic means $\overline{U}_{1N}$ and $\overline{U}_{1X}$, the nominal values of the standard resistor $R_{1N}$ and the internal resistance of voltmeter $R_{1V}$, i.e.,

$$R_{1X} = f\left(R_{1N}, R_{1V}, \overline{U}_{1N}, \overline{U}_{1X}\right) \tag{26}$$

where $\overline{U}_{1N}$ and $\overline{U}_{1X}$ are obtained by means of (11) and (13), respectively.

The output estimate $R_{1X}$ is taken as the arithmetic mean or average of $n$ independent determinations $R_{1X,k}$ ($k = 1, \ldots, n$) of $R_X$, each determination having the same uncertainty and each being based on a complete set of observed values of the four input estimates $R_{1N}$, $R_{1V}$, $U_{1N}$, and $U_{1X}$ that are obtained simultaneously, i.e.,

$$R_{1X} = \overline{R}_X = \frac{1}{n}\sum_{k=1}^{n} R_{X,k} = \frac{1}{n}\sum_{k=1}^{n} f\left(R_{N,k}, R_{V,k}, U_{N,k}, U_{X,k}\right) \tag{27}$$

The output estimate $R_{1X}$ is obtained by means of (27) may be preferable when the measurement model function $f(\cdot)$ is a nonlinear function, but the two approaches are identical if $f(\cdot)$ is a linear function of the input quantities (provided that the experimentally observed correlation coefficients are taken into account when implementing the first approach). In the measurement practice, the value of the uncertainty of the measurement is generally low with respect to the measured value, hence it determines small variations of the measurand. This means that the linearity condition of $f(\cdot)$ is almost always locally verified, near the measurement point.

Presuming that the function $f(\cdot)$ in (10) is fairly linear, about the measured value $R_{1X}$, at least for small deviations of each of the four input quantities $R_N$, $R_V$, $U_N$, and $U_X$, about their estimates $R_{1N}$, $R_{1V}$, $U_{1N}$, and $U_{1X}$, respectively. When the input quantities $U_{1N,k}$ and $U_{1X,k}$ ($k = 1, \ldots, n$) are correlated in the repeated observations assuming that that function $f(\cdot)$ in (10) is fairly linear, about the measured value $R_{1X}$, then the determination of the output estimate $R_{1X}$ by means of (26) will be more convenient for further analysis.

The standard uncertainty of the result of the measurement $R_{1X}$, which is obtained from the values of the four input quantities $R_N$, $R_V$, $U_N$, and $U_X$, is termed a combined standard uncertainty and denoted by $u_C(R_{1X})$. In the general case, with the correlated estimates of the input values and assuming that the measurement model function $f(\cdot)$ is a linear function, the combined standard uncertainty $u_C(R_{1X})$ of the measurement result $R_{1X}$ is the estimated standard deviation that is associated with the result $R_{1X}$ and is equal to the positive square root of the combined variance $u_C^2(R_{1X})$ that is obtained from the following variance and covariance components

$$u_C^2(R_{1X}) \quad = \left(\frac{\partial f}{\partial R_{1N}}\right)^2 u_B^2(R_{1N}) + \left(\frac{\partial f}{\partial R_{1V}}\right)^2 u_B^2(R_{1V}) + \left(\frac{\partial f}{\partial \overline{U}_{1X}}\right)^2 u^2(\overline{U}_{1X}) + \left(\frac{\partial f}{\partial \overline{U}_{1N}}\right)^2 u^2(\overline{U}_{1N}) +$$

$$+ 2\frac{\partial f}{\partial \overline{U}_{1N}}\frac{\partial f}{\partial \overline{U}_{1X}}u(\overline{U}_{1N},\overline{U}_{1X}) + 2\frac{\partial f}{\partial R_{1N}}\frac{\partial f}{\partial R_{1V}}u(R_{1N},R_{1V}) +$$

$$+ 2\frac{\partial f}{\partial \overline{U}_{1N}}\frac{\partial f}{\partial R_{1N}}u(\overline{U}_{1N},R_{1N}) + 2\frac{\partial f}{\partial \overline{U}_{1N}}\frac{\partial f}{\partial R_{1V}}u(\overline{U}_{1N},R_{1V}) +$$

$$+ 2\frac{\partial f}{\partial \overline{U}_{1X}}\frac{\partial f}{\partial R_{1N}}u(\overline{U}_{1X},R_{1N}) + 2\frac{\partial f}{\partial \overline{U}_{1X}}\frac{\partial f}{\partial R_{1V}}u(\overline{U}_{1X},R_{1V})$$

(28)

where

$u_B(R_{1N})$—Type B evaluation of standard uncertainty that is associated with the input estimate $R_{1N}$, which can be obtained by means of (22),

$u_B(R_{1V})$—Type B evaluation of standard uncertainty that is associated with the input estimate $R_{1V}$, which can be obtained by means of (22),

$u(\overline{U}_{1N})$—may be Type A or either Type B evaluation of standard uncertainty that is associated with the input estimate $U_{1N}$. Type A evaluation can be obtained by means of (17) and Type B evaluation can be obtained by means of (22),

$u(\overline{U}_{1X})$—may be Type A or either Type B evaluation of standard uncertainty that is associated with the input estimate $U_{1X}$. Type A evaluation can be obtained by means of (18) and Type B evaluation can be obtained by means of (22),

$u(\overline{U}_{1N},\overline{U}_{1X})$—estimate of the covariance of input means $\overline{U}_{1N}$ and $\overline{U}_{1X}$,

$u(R_{1N},R_{1V})$—estimate of the covariance of input estimates $R_{1N}$ and $R_{1V}$,

$u(\overline{U}_{1N},R_{1N})$—estimate of the covariance of input mean $\overline{U}_{1N}$ and input estimate $R_{1N}$,

$u(\overline{U}_{1N},R_{1V})$—estimate of the covariance of input mean $\overline{U}_{1N}$ and input estimate $R_{1V}$,

$u(\overline{U}_{1X},R_{1N})$—estimate of the covariance of input mean $\overline{U}_{1X}$ and input estimate $R_{1N}$,

$u(\overline{U}_{1X},R_{1V})$—estimate of the covariance of input mean $\overline{U}_{1X}$ and input estimate $R_{1V}$.

The partial derivatives $\frac{\partial f}{\partial R_{1N}}$, $\frac{\partial f}{\partial R_{1V}}$, $\frac{\partial f}{\partial \overline{U}_{1N}}$ and $\frac{\partial f}{\partial \overline{U}_{1X}}$ (often referred to as sensitivity coefficients) are equal to $\frac{\partial f}{\partial R_N}$, $\frac{\partial f}{\partial R_V}$, $\frac{\partial f}{\partial U_N}$ and $\frac{\partial f}{\partial U_X}$ evaluated at $R_N = R_{1N}$, $R_V = R_{1V}$, $U_N = \overline{U}_{1N}$, and $U_X = \overline{U}_{1X}$, respectively.

Equation (28) is based on the first-order Taylor series approximation of the model function of the measurement $R_X = f(R_N, R_V, U_N, U_X)$ and it expresses what is termed in the Guide [1] as the law of propagation of uncertainty.

It is noted in [1] that the covariance that is associated with the estimates of two input quantities may be taken to be zero or treated as insignificant if

- these input quantities are uncorrelated,
- either of these two input quantities can be treated as a constant, or if
- there is insufficient information to evaluate the covariance that is associated with the estimates of these two input quantities.

Hence, considering that the estimates $R_{1N}$ and $R_{1V}$ are constant during the measurement by the proposed method, then the estimated covariances $u(R_{1N}, R_{1V})$, $u(\overline{U}_{1N}, R_{1N})$, $u(\overline{U}_{1N}, R_{1V})$, $u(\overline{U}_{1X}, R_{1N})$, and $u(\overline{U}_{1X}, R_{1V})$ may be ignored in (28). Since it frequently occurs that no associated uncertainty $u_B(R_{1V})$ is stated in the manufacturer's specification of the voltmeter, it can also be ignored in (28).

In accordance with this explanation, Equation (28) is reduced and may be rendered as follows

$$u_C^2(R_{1X}) \quad = \left(\frac{\partial f}{\partial R_{1N}}\right)^2 u_B^2(R_{1N}) + \left(\frac{\partial f}{\partial \overline{U}_{1X}}\right)^2 u^2(\overline{U}_{1X}) +$$

$$+ \left(\frac{\partial f}{\partial \overline{U}_{1N}}\right)^2 u^2(\overline{U}_{1N}) + 2\frac{\partial f}{\partial \overline{U}_{1N}}\frac{\partial f}{\partial \overline{U}_{1X}}u(\overline{U}_{1N},\overline{U}_{1X})$$

(29)

The partial derivatives or sensitivity coefficients in (29) are evaluated by means of the following expressions:

$$\frac{\partial f}{\partial R_{1N}} = \frac{\overline{U}_{1N}}{\overline{U}_{1X}} \cdot \frac{1}{R_{1N}^2} \left[ \frac{\overline{U}_{1N}}{\overline{U}_{1X}} \left( \frac{1}{R_{1N}} + \frac{1}{R_{1V}} \right) - \frac{1}{R_{1V}} \right]^{-2} \tag{30}$$

$$\frac{\partial f}{\partial \overline{U}_{1X}} = \frac{\overline{U}_{1N}}{\overline{U}_{1X}^2} \left( \frac{1}{R_{1N}} + \frac{1}{R_{1V}} \right) \left[ \frac{\overline{U}_{1N}}{\overline{U}_{1X}} \left( \frac{1}{R_{1N}} + \frac{1}{R_{1V}} \right) - \frac{1}{R_{1V}} \right]^{-2} \tag{31}$$

and

$$\frac{\partial f}{\partial \overline{U}_{1N}} = -\frac{1}{\overline{U}_{1X}} \left( \frac{1}{R_{1N}} + \frac{1}{R_{1V}} \right) \left[ \frac{\overline{U}_{1N}}{\overline{U}_{1X}} \left( \frac{1}{R_{1N}} + \frac{1}{R_{1V}} \right) - \frac{1}{R_{1V}} \right]^{-2}. \tag{32}$$

where $\overline{U}_{1N}$ and $\overline{U}_{1X}$ are obtained by means of (11) and (13), respectively.

The terms $u(\overline{U}_{1N}, \overline{U}_{1X})$ in (28) represent the estimate of the covariance of input means $\overline{U}_{1N}$ and $\overline{U}_{1X}$, determined from $n$ independent pairs of repeated simultaneous observations $U_{1N,k}$ and $U_{1X,k}$ ($k = 1, \ldots, n$) of the estimates $U_{1N}$ and $U_{1X}$. If there are $n$ pairs of measured results of independent repeated measurements of the estimates $U_{1N}$ and $U_{1X}$, then the covariance $u(\overline{U}_{1N}, \overline{U}_{1X})$ as a statistical measurement of the strength of the correlation between these $n$ pairs of the estimates can be calculated by the following equation:

$$u(\overline{U}_{1N}, \overline{U}_{1X}) = u(\overline{U}_{1X}, \overline{U}_{1N}) = \frac{1}{N(N-1)} \sum_{k=1}^{N} (U_{1N,k} - \overline{U}_{1N})(U_{1X,k} - \overline{U}_{1X}) \tag{33}$$

The degree of correlation between $u(\overline{U}_{1N})$ and $u(\overline{U}_{1X})$ is characterized by the estimated correlation coefficient of input means $\overline{U}_{1N}$ and $\overline{U}_{1X}$, defined as

$$r(\overline{U}_{1N}, \overline{U}_{1X}) = r(\overline{U}_{1X}, \overline{U}_{1N}) = \frac{u(\overline{U}_{1N}, \overline{U}_{1X})}{u(\overline{U}_{1N}) u(\overline{U}_{1X})} \quad -1 \leq r(\overline{U}_{1N}, \overline{U}_{1X}) \leq 1 \tag{34}$$

In terms of the correlation coefficients, which are more readily interpreted than covariances, and by using (34), Equation (29) may be written as

$$\begin{aligned} u_C^2(R_{1X}) &= \left( \frac{\partial f}{\partial R_{1N}} \right)^2 u_B^2(R_{1N}) + \left( \frac{\partial f}{\partial \overline{U}_{1X}} \right)^2 u^2(\overline{U}_{1X}) + \left( \frac{\partial f}{\partial \overline{U}_{1N}} \right)^2 u^2(\overline{U}_{1N}) + \\ &+ 2 \frac{\partial f}{\partial \overline{U}_{1N}} \frac{\partial f}{\partial \overline{U}_{1X}} u(\overline{U}_{1N}) u(\overline{U}_{1X}) r(\overline{U}_{1N}, \overline{U}_{1X}) \end{aligned} \tag{35}$$

which is also known as the general formulation of the law of propagation of uncertainty [1]. With the uncorrelated estimates of the input values, $r(\overline{U}_{1N}, \overline{U}_{1X}) = 0$, and consequently

$$u_C^2(R_{1X}) = \left( \frac{\partial f}{\partial R_{1N}} \right)^2 u_B^2(R_{1N}) + \left( \frac{\partial f}{\partial \overline{U}_{1X}} \right)^2 u^2(\overline{U}_{1X}) + \left( \frac{\partial f}{\partial \overline{U}_{1N}} \right)^2 u^2(\overline{U}_{1N}) \tag{36}$$

The uncertainty propagation law (35) (or its simplified version (36) in case of the uncorrelated input quantities) can be used to evaluate the combined standard uncertainty $u_C(R_{1X})$ of the result of the unknown resistance measurement when the measurand $R_X$ is not measured directly, but is determined from the four input quantities $R_N$, $R_V$, $U_N$, and $U_X$ according to (8), while the standard uncertainties of their estimates are known. The combined standard uncertainty $u_C(R_{1X})$ is an estimated standard deviation and it indicates the dispersion of the values that are reasonably attributable to the measurand $R_X$. The resulting combined standard uncertainty can be used to obtain an expanded uncertainty with a provided coverage probability.

*2.4. Evaluation of Output Quantity and Its Uncertainty Considering Thermo-Electrical Voltages*

Thermoelectric voltages can seriously affect the low resistance measurement accuracy. The current reversal method, the delta method, and the offset-compensated ohms method are three common ways to overcome these unwanted offsets. When the thermoelectric voltages are constant with respect to the measurement cycle, the current-reversal method will successfully compensate for these offsets. However, if the changing thermoelectric voltages are causing inaccurate results, then the delta method should be used. The delta method is similar to the current reversal method in terms of alternating the current source polarity, but it differs in using three voltage measurements to perform each resistance calculation. The current reversal method provides a twice better signal-to-noise ratio and, therefore, better accuracy than the offset-compensated ohms method. Hence, in this paper we will use the current reversal method to cancel the thermoelectric voltage, which is also used in [19].

Thermoelectric voltages can be cancelled by making two measurements with the currents of opposite polarity, as shown in Figure 3b. This can be achieved by measuring the voltages $U_N$ and $U_X$ for both polarities of the power supply (measuring first the voltage $U_N$ for both polarities, and afterwards the voltage $U_X$ for both polarities). The averaging of the voltage measurement results by both polarities of the power supply, i.e.,

$$U_{1N} = \frac{U_{1N+} - U_{1N-}}{2} \tag{37}$$

and

$$U_{1X} = \frac{U_{1X+} - U_{1X-}}{2}, \tag{38}$$

allows the elimination of the influence of thermo-electrical voltages. In the case of several repeated voltage measurements, substituting the voltage results from (37) and (38) into (11) and (13), and then into (8), will provide an estimation of the unknown resistance $R_{1X}$. In the case of a single observation of the voltages $U_N$ and $U_X$, the results from (37) and (38) are immediately substituted in (8). It is noted that the thermoelectric voltages are completely cancelled out by this approach.

Type A evaluation of standard uncertainty $u(\overline{U}_{1N})$ can be obtained by means of (17), and Type A evaluation of standard uncertainty $u(\overline{U}_{1X})$ can be obtained by means of (18).

In the case of a single observation of the voltages $U_N$ and $U_X$ and according to [19], it can be assumed that there is $u_C(R_{1X+}) \doteq u_C(R_{1X-})$, and it, therefore, suffices to estimate only one of these uncertainties, i.e., the uncertainty of the measured resistance $R_{1X}$ for one power supply polarity only. The resulting combined standard uncertainty of the averaged value $R_{1X} = (R_{1X+} + R_{1X-})/2$ can be derived from the equation

$$u_C(R_{1X}) = \sqrt{\left(\frac{u_C(R_{1X+})}{2}\right)^2 + \left(\frac{u_C(R_{1X-})}{2}\right)^2} = \sqrt{\frac{2u_C^2(R_{1X+})}{4}} = \frac{u_C(R_{1X+})}{\sqrt{2}} \tag{39}$$

where $u_C(R_{1X+})$ and $u_C(R_{1X-})$ can be obtained by means of (36).

*2.5. Determining Expanded Uncertainty*

The combined standard uncertainty of the measurement result $u_C(R_{1X})$ defines an interval $R_{1X} - u_C(R_{1X})$ to $R_{1X} + u_C(R_{1X})$ about the measurement result $R_{1X}$ within which the value of the measurand $R_X$ estimated by $R_{1X}$ can be confidently asserted to lie to the extent of 68% for the normal (Gaussian) probability distribution or to the extent of approximately 57.7% for the rectangular probability distribution. That is, it is confidently believed that

$R_{1X} - u_C(R_{1X}) \leq R_X \leq R_{1X} + u_C(R_{1X})$ which is commonly written as $R_X = R_{1X} \pm u_C(R_{1X})$.

In many areas of the industrial measuring practice, a coverage probability of $p = 68.3\%$ is found to be too low. It is the intention of this paper to provide an interval about the

result of a measurement $R_{1X}$ that may be expected to encompass a large fraction of the distribution of values that could reasonably be attributed to the measurand $R_X$. Hence, the expanded uncertainty is introduced in this analysis and its value is given by

$$U = k \cdot u_C(R_{1X}) \tag{40}$$

where $k$ is a coverage factor. In terms of the rectangular distribution, the value of the factor $k$ usually ranges from 1.5 to 1.73 and it is based on the coverage probability, or the level of confidence that is required from the interval, Table 1. The relationship between the coverage factor $k$ and the coverage probability $p$ for the rectangular distribution is given by

$$k = p\sqrt{3} \tag{41}$$

**Table 1.** Values of the coverage factor $k$ that produces an interval with the coverage probability $p$ assuming a rectangular distribution.

| Coverage Probability $p$ | Coverage Factor $k$ |
|:---:|:---:|
| 0.90 | 1.559 |
| 0.95 | 1.645 |
| 0.99 | 1.715 |
| 1.00 | 1.732 |

For the normal (Gaussian) probability distribution, the value of the factor $k$ is usually in the range from one to three and it is based on the coverage probability, or the level of confidence required for the interval, Table 2.

**Table 2.** Values of the coverage factor $k$ that produces an interval with the coverage probability $p$ assuming a normal distribution.

| Coverage Probability $p$ | Coverage Factor $k$ |
|:---:|:---:|
| 0.6827 | 1 |
| 0.90 | 1.645 |
| 0.95 | 1.96 |
| 0.9545 | 2 |
| 0.99 | 2.576 |
| 0.9973 | 3 |

Finally, the value of the measured unknown resistance can be expressed by using the expanded uncertainty by means of the following expression

$$R_X = R_{1X} \pm k \cdot u_C(R_{1X}) = R_{1X} \pm U \tag{42}$$

## 3. Practical Example

This practical example demonstrates both approaches for determining the value of the unknown electrical resistance by applying the direct comparison method using a standard resistor and voltmeter and by evaluating its combined standard uncertainty. The unknown resistor $R_X$, the standard resistor $R_N$ and the current source are connected in a series, as shown in Figure 1, and a voltmeter (DMM) is used to measure the voltage drop across each resistor. All the connections according to Figure 1 are made by the same type of conductor and thus the amount of thermoelectric EMF that was added to the voltage measurement will be negligible.

Measuring instruments and devices with the following data were used:

1.　Single-channel laboratory linear DC power supply, Model GPS-3030DD (30 V/3 A single output, 90 W), GW Instek,
2.　Variable resistor PRN 533 (as the unknown resistor $R_X$) with the rated resistance ranging from 0 to1 kΩ and the continuous permitted current 0.57 A. Resistance tolerance was +10%,

3. Resistance decade MA 2125 (as the standard resistor $R_N$) with the standard resistance ranging from 0 to 9,999999 MΩ. The accuracy class of this resistance decade was ±1% + 0.08 Ω. The maximum current rating ranging from 0 to 999 Ω was 25 mA,

4. TRMS multi-meter, Model EX542, EXTECH (as the voltmeter) with 40,000 count LCD display and input impedance >10 MΩ VDC. The accuracy of this multi-meter for DC voltage was ±(0.06% reading + 4 digits) with resolution of 0.0001 V.

During the measurement that was performed in line with the proposed model, the slider of the variable resistor PRN 533 was set to an arbitrary position. The desired value of the resistance decade MA 2125 was predicted and set to 240 Ω. The current intensity through the resistors $R_X$ and $R_N$ during the measurement was constantly maintained at 10 mA, which is considerably less than the continuous permissible currents through these resistors, and at the same time this amount of current intensity will not cause a noticeable temperature rise of these resistors, hence their electrical resistances can be considered constant during the measurement.

Since the internal resistance of the voltmeters is very large (>10 MΩ VDC) in relation to the measured electrical resistance, and the conditions (a) and (b) of Chapter 2.1 are also fulfilled, then the measurand (output quantity) $R_X$ may be determined from the three input quantities $R_N$, $U_N$, and $U_X$ through the functional relationship $f(\cdot)$ that is given by means of (7). Hence, the combined standard uncertainty $u_C(R_{1X})$ can be obtained in the case of correlated input quantities by means of (35).

Since the conditions (a), (b), and (c) of Chapter 2.1 are fulfilled in this measurement, then the output estimate $R_{1X}$ is calculated by means of (2) according to both proposed approaches.

In this example, we considered eleven independent sets of simultaneous observations of two input quantities $U_N$ and $U_X$ that were obtained under similar conditions, resulting in the data that is provided in Table 3. The arithmetic means of the observations and the experimental standard deviations of those means that were calculated from Equations (11), (13), (17) and (18) are also given.

According to the first approach, the means are taken as the optimum estimates of the expected input quantity values, and the experimental standard deviations represent the standard uncertainties of those means.

Table 4 shows the measured input data and the analysis results in accordance with the second approach. By comparing the results of the calculations from Tables 3 and 4, it is noted that the same values for resistance $R_{1X}$ were obtained by using both approaches and that the combined standard uncertainty $u_C(R_{1X})$ that was obtained from the first approach was less than 1.39% compared to the amount of $u_C(R_{1X})$ that was obtained in the second approach. These results justify the introduction of the assumption into the proposed model that the function $f(\cdot)$ given by (7) or by (10) is a linear function.

**Table 3.** Value of the output quantity $R_X$ and its combined uncertainty $u_C$ that were obtained according to the first approach from eleven sets of simultaneous observations.

| Set Number $k$ | Input Quantities | | | Output Quantity $R_{1X}$ ($\Omega$) | Combined Standard Uncertainty $u_C(R_{1X})$ [$\Omega$] |
| --- | --- | --- | --- | --- | --- |
| | $U_{1X}$ (V) | $U_{1N}$ (V) | $R_{1N}$ ($\Omega$) | | |
| 1 | 1.1418 | 4.636 | 240 | | |
| 2 | 1.1418 | 4.635 | 240 | | |
| 3 | 1.1313 | 4.594 | 240 | | |
| 4 | 1.1315 | 4.594 | 240 | | |
| 5 | 1.133 | 4.601 | 240 | | |
| 6 | 1.1331 | 4.601 | 240 | | |
| 7 | 1.1332 | 4.601 | 240 | | |
| 8 | 1.1325 | 4.598 | 240 | | |
| 9 | 1.1325 | 4.598 | 240 | | |
| 10 | 1.1355 | 4.611 | 240 | | |
| 11 | 1.1356 | 4.611 | 240 | | |
| Arithmetic mean | 1.134709 | 4.6072727 | 240 | | |
| Experimental standard deviation of mean | 0.001134 | 0.0045369 | | | |
| $u_B(R_{1N})$ | | | 1.431828668 | | |
| $R_{1X} = \overline{R}_{1N} \cdot \overline{U}_{1X}/\overline{U}_{1N}$ | | | | 59.10876085 | |
| $u(\overline{U}_{1N}, \overline{U}_{1X})$ | $5.14248 \times 10^{-6}$ | | | | |
| $\frac{\partial f}{\partial R_{1N}} = \frac{\overline{U}_{1X}}{\overline{U}_{1N}}$ | 0.246286504 | | | | |
| $\frac{\partial f}{\partial \overline{U}_{1X}} = \frac{R_{1N}}{\overline{U}_{1N}}$ | | | 52.09155485 | | |
| $\frac{\partial f}{\partial \overline{U}_{1N}} = \frac{-R_{1N} \cdot \overline{U}_{1X}}{\overline{U}_{1N}^2}$ | | −12.82944691 | | | |
| $u_C(R_{1X})[\Omega]$ | | | | | 0.352645412 |

**Table 4.** Value of the output quantity $R_X$ and the associated combined uncertainty $u_C$ that was obtained according to the 2nd approach from eleven sets of simultaneous observations.

| Set Number $k$ | Input Quantities | | | Output Quantity $R_{1X}$ ($\Omega$) | $ua(U_{1X})$ [V] | $u_B(U_{1X})$ [V] | $ua(U_{1N})$ [V] | $u_B(U_{1N})$ [V] | $u_C(R_{1X})$ [$\Omega$] |
| --- | --- | --- | --- | --- | --- | --- | --- | --- | --- |
| | $U_{1X}$ (V) | $U_{1N}$ (V) | $R_{1N}$ ($\Omega$) | | | | | | |
| 1 | 1.1418 | 4.636 | 240 | 59.10957722 | 0.0010851 | 0.000626471 | 0.0067816 | 0.003915359 | 0.357691535 |
| 2 | 1.1418 | 4.635 | 240 | 59.1223301 | 0.0010851 | 0.000626471 | 0.006781 | 0.003915012 | 0.357690911 |
| 3 | 1.1313 | 4.594 | 240 | 59.10143666 | 0.0010788 | 0.000622834 | 0.0067564 | 0.003900809 | 0.357648132 |
| 4 | 1.1315 | 4.594 | 240 | 59.11188507 | 0.0010789 | 0.000622903 | 0.0067564 | 0.003900809 | 0.357648459 |
| 5 | 1.133 | 4.601 | 240 | 59.10019561 | 0.0010798 | 0.000623423 | 0.0067606 | 0.003903234 | 0.357655271 |
| 6 | 1.1331 | 4.601 | 240 | 59.10541187 | 0.0010799 | 0.000623457 | 0.0067606 | 0.003903234 | 0.357655434 |
| 7 | 1.1332 | 4.601 | 240 | 59.11062812 | 0.0010799 | 0.000623492 | 0.0067606 | 0.003903234 | 0.357655598 |
| 8 | 1.1325 | 4.598 | 240 | 59.11265768 | 0.0010795 | 0.00062325 | 0.0067588 | 0.003902195 | 0.357652585 |
| 9 | 1.1325 | 4.598 | 240 | 59.11265768 | 0.0010795 | 0.00062325 | 0.0067588 | 0.003902195 | 0.357652585 |
| 10 | 1.1355 | 4.611 | 240 | 59.10214704 | 0.0010813 | 0.000624289 | 0.0067666 | 0.003906698 | 0.357665595 |
| 11 | 1.1356 | 4.611 | 240 | 59.10735198 | 0.0010814 | 0.000624323 | 0.0067666 | 0.003906698 | 0.357665759 |
| Arithmetic mean | 1.13471 | 4.60727 | 240 | 59.10875264 | | | | | 0.357661988 |

Both approaches for calculating the output estimate $R_{1X}$ and its associated combined standard uncertainty $u_C(R_{1X})$ are carried out by means of the Microsoft Excel software, while Tables 3 and 4 represent parts of the corresponding Microsoft Excel worksheet.

Figure 4 presents the results of the analysis that were carried out by means of the GUM Workbench Edu software [20] using the same input data.

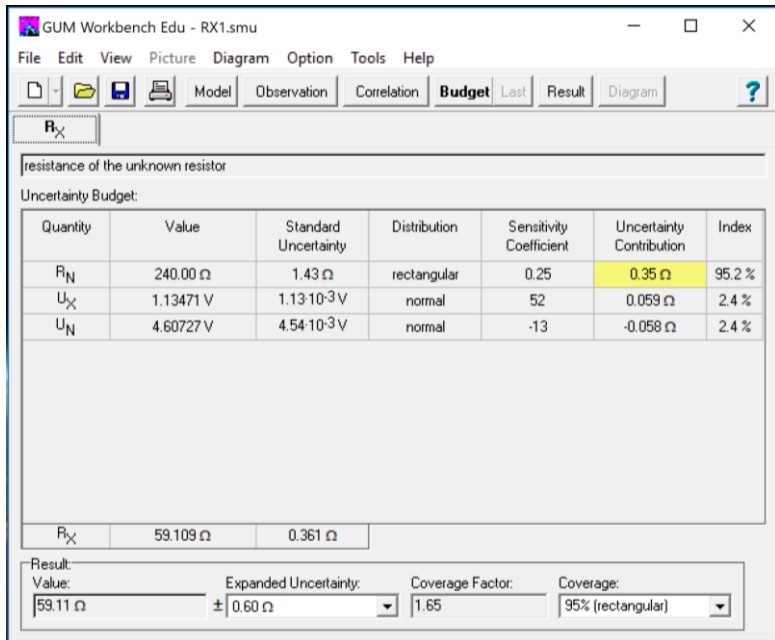

**Figure 4.** Value of the output quantity $R_X$ and its combined uncertainty $u_C$ that was obtained by using the GUM Workbench Edu from eleven sets of simultaneous observations.

Table 5 gives a comparative presentation of the results for $R_{1X}$ and $u_C(R_{1X})$ from Tables 3 and 4, and from Table in Figure 4.

**Table 5.** Comparative presentation of the results for $R_{1X}$ and $u_C(R_{1X})$.

|  | Table 3 | Table 4 | Figure 4 |
|---|---|---|---|
| $R_{1X}$ ($\Omega$) | 59.10876085 | 59.10875264 | 59.11 |
| $u_C(R_{1X})$ ($\Omega$) | 0.352645412 | 0.357661988 | 0.361 |

Table 5 shows good congruence results for $R_{1X}$ and $u_C(R_{1X})$ from Table in Figure 4 with corresponding results given in Tables 3 and 4.

The uncertainty budget from Figure 4 shows that the standard uncertainty $u_B(R_{1N})$ contributes the most (95.2%) to the combined standard $u_C(R_{1X})$. Hence, it can be concluded that the combined standard uncertainty $u_C(R_{1X})$ is limited by the precision and accuracy of the standard resistor $R_{1N}$.

It is very important to note that additional measurements were conducted with set values 140 $\Omega$ and 940 $\Omega$ of the resistance decade MA 2125 and that very similar results were obtained to abovementioned results.

## 4. Conclusions

In this paper, the mathematical expression (8) consists of the old expression which is modified to represent the new and the original form for the calculation of unknown electrical resistance. This new form is extremely suitable for conducting a partial derivation which can be of great importance if one assumes that the model of measurement functions has a nonlinear character. This paper sets out three conditions that must be met in order to minimize and ignore the systematic measurement deviations. The fulfilment of these three conditions allows the use of the theoretical expression (2) for the calculation of unknown electrical resistance. The complete model function of the measurement that is given by (10), or a simplified model function given by (7) are assumed to have functional linear dependence. The results that were obtained from the illustrative example justify the introduction of such assumption.

In this paper, two approaches for the estimation of unknown resistance $R_X$ and its combined standard uncertainty are analyzed. The set of input quantities $R_N$, $R_V$, $U_N$, and

$U_X$ is categorized in this paper as the quantities whose estimated values and uncertainties are obtained from manufacturer's specification ($R_N$ and $R_V$) and from a single observation or from repeated observations ($U_N$ and $U_X$). The results that were obtained in the illustrative example indicate that it is not necessary to perform repeated voltage measurements, i.e., only one measurement suffices because the standard uncertainty $u_B(R_{1N})$ that is associated with the standard resistor contributes the most (95.2%) to the combined standard $u_C(R_{1X})$. It is very important to note that the combined standard uncertainty is limited by the precision and the accuracy of the standard resistance $R_N$. The mathematical apparatus for the statistical analysis of this indirect measuremen has been substantially elaborated.

The method that is proposed in this paper allows the elimination of the influence of thermo-electrical voltages, if any.

For many purposes, the unknown resistors are compared to standard resistors by a comparison circuit (comparators). In this method, it is possible to use the proposed approach for the estimation of the combined standard uncertainty that is associated with the measurand too. Analogously, it is also possible to determine the unknown electrical resistance and its associated combined standard uncertainty by the measurement method that is referred to as the direct comparison method using a standard resistor and potentiometer [21].

This paper has practical application and can be used for educational purposes.

**Author Contributions:** Conceptualization, V.P. and V.B.; methodology, V.P. and V.B.; software, V.P. and V.B.; validation, V.P., V.B. and H.T.; formal analysis, V.P. and V.B.; investigation, V.P., V.B. and H.T.; resources, V.P., V.B. and H.T.; data curation, V.P., V.B. and H.T.; writing—original draft preparation, V.P. and V.B.; writing—review and editing, V.P.; visualization, V.B.; supervision, V.B.; project administration, V.P.; funding acquisition, V.P. All authors have read and agreed to the published version of the manuscript.

**Funding:** This research received no external funding.

**Institutional Review Board Statement:** Not applicable.

**Informed Consent Statement:** Not applicable.

**Data Availability Statement:** Not applicable.

**Conflicts of Interest:** The authors declare no conflict of interest.

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
