# Peer review of "The Measurement Uncertainty in Determining of Electrical Resistance Value by Applying Direct-Comparison Method"

_energies, doi:10.3390/en15062115_

Round 1
Reviewer 1 Report
The manuscript entitled 'Uncertainty of measurement at the determining of unknown electrical resistance by applying direct-comparison measurement method using a standard resistor and voltmeter' has been completely reviewed. I have some suggestions that may help to improve the paper quality further:
1- The 'Title' of the article is unreasonably long. In my opinion, it should be concise and revised based on the work.
2- Also, 'Introduction' is not well focused. In a research paper, it is expected that introduction section briefly explains the starting background and, even more important, the originality and novelty and relevancy of the study is well established. In this case, authors do not put effort enough in highlighting the relevancy and the novelty of the study. So that most of references 1 and 2 have been researched and used.
3- Why mathematical symbols and signs in equations 3, 4 and 5 are not clear in PDF file?
4- The results do not show well the volume of content and equations presented. It is better to present the percentage of changes obtained from the results for power, voltage and current responses to R in the abstract.
5- It is suggested to compare the obtained values of the output quantity R and its combined uncertainty u with another similar system.
Author Response
Dear Sir,
thank you for your review. Please, see the attachment with response to comments.
best regards

Reviewer 2 Report
The paper presents the problem of determining uncertainty of resistance measurement by indirect method, using a standard resistor and a voltmeter. Measurement systems for two- and four-wire connection system are presented. The equation of measurement is proposed and the individual standard uncertainties are described. The final part of the paper presents a practical example of determining the resistance value and the uncertainty budget for a specific measurement system for two approaches.
The considered issue is interesting especially from the practical point of view.
The paper has medium scientific level and is written in fairly good language, but some of the sentences are too long. Therefore some parts of this paper are difficult to understand. I recommend to take this fact into account in future publications.
The list of References is sufficient.
Below, a few remarks were presented. Please take them into consideration in final version of this article.
- Please change the affiliation number for the third author.
- There is a rectangle symbol in equations 3, 4, 5 and in line 484. Please change it.
- Equation (15) defines the standard deviation and not the Type A uncertainty. The correct formula for Type A uncertainty is given by (17). Similarly, (16) and (18).
- The symbol a, formulas (23) and (24) indicates the absolute error value. Please explain the ND component in formula (24). In the description of the formula for the error value of the EX542 multimeter, line 548, this component does not appear.
- In lines 355 and 405, please correct the entries
- In Table 4, in columns 6 and 8, insert the uncertainty symbol of type A, uA instead of a.
Author Response

(The authors gave the same response as above.)

Round 2
Reviewer 1 Report
Thank you for your response and clarifications.